# Sexual activity and contraceptive use among adolescents: A descriptive survey in a Ghanaian municipality

Peter Boakye *, Evans Adaboh, Alberta Yemotsoo Lomotey, Jacob Tetteh,
Abigail Kusi-Amponsah Diji, Victoria Bubunyo Bam

School of Nursing and Midwifery, College of Health Sciences, Kwame Nkrumah University of Science and Technology, Kumasi, Ghana

* peterboakye180@gmail.com

## Abstract

Contraceptive use is essential for reducing unwanted pregnancies and sexually transmitted infections (STIs) among adolescents. Sexual activity during adolescence, particularly in developing countries, remains a major public health concern with significant implications for reproductive health. Adolescents aged 10–19 undergo significant hormonal changes that contribute to heightened sexual drive and an increased likelihood of early sexual activity. However, despite widespread knowledge of modern contraceptive methods, their actual use remains low. This study aims to assess sexual activity and contraceptive use among Senior High School (SHS) adolescents and to identify key predictors of contraceptive uptake. A cross-sectional study was conducted from August to September 2022 involved 330 adolescents selected through simple random sampling. Data were analyzed using frequencies, percentages, Pearson's chi-squared test, and binary logistic regression to examine associations between sociodemographic characteristics (age, sex, class, ethnicity, religion, residential status), sexual activity characteristics (relationship status, multiple sexual partners), awareness of contraceptives, and contraceptive use. Statistical significance was set at p ≤ 0.05 with a 95% confidence interval. Out of the total participants, 290 (87.9%) were within the age range of 15–19 years and the majority were females (n = 196, 59.4%). Approximately half (n = 166, 50.3%) were in an intimate relationship and 126 (38%) had engaged in sexual intercourse. Among sexually active adolescents, over half (n = 65, 51.6%) had their first sexual intercourse between the ages of 10–14 years and 100 (79.4%) reported having ever used contraceptive, primarily condoms (n = 66, 66.0%), and pills (n = 43, 43.0%), with IUDs being the least common (n = 2, 2.0%). Sexually active adolescents who were aware of contraceptive (AOR = 6.686, 95%CI = 1.515 – 29.505, p = 0.012) had higher odds of contraceptive use. Early sexual initiation and contraceptive use are prevalent among adolescents. Comprehensive sex education and peer counseling on reproductive health are

**Data availability statement:** All relevant data are within the paper and its Supporting Information files.

**Funding:** The author(s) received no specific funding for this work.

**Competing interests:** The authors have declared that no competing interests exist.

needed to address the consequences of early sexual activity without contraceptive use.

## 1. Introduction

Adolescent sexual activity is a global public health concern that significantly impacts reproductive health outcomes [1], especially in developing countries such as Ghana [2]. Globally, more than 16 million adolescent girls aged 15–19 years give birth each year, most of which occur in Low- and Middle-Income Countries (LMICs) [3,4]. Adolescence is a transitional phase of growth and development between childhood and adulthood between the ages of 10–19 [5]. This phase is a very critical period for sexual drive due to hormonal changes that occur naturally at this stage of life [6]. In Ghana, as in many other countries globally, adolescents engage in sexual activity at earlier ages, often initiating sexual intercourse before the age of 15 [7,8], contributing to a host of challenges, including unintended pregnancies, unsafe abortions, and sexually transmitted infections (STIs) [7,9].

According to the World Health Organization (WHO) [5], 37.1% of female adolescents and 21.3% of male adolescents had ever had sex in Ghana. Among these, 9.9% of females and 6.9% of males reported having sex before the age of 15. In 2020, the adolescent birth rate was 62.5 per 1,000 adolescent girls, and nearly 20,000 adolescents were living with HIV, with more than 2,000 new cases reported annually [5].

The World Health Organization emphasizes that adolescents have the right to access appropriate information and services regarding sexual and reproductive health [10]. However, a lack of comprehensive knowledge, misconceptions, fears, and social and cultural norms, including religious prejudices limit adolescents' contraceptive use [11]. Additionally, psychological factors, such as fear of judgement from peers and parents, fear of side effects, and accessibility issues, like the location of service points, make adolescents hesitant to discuss or seek contraceptive options [12,13]. Consequently, many adolescents engage in unprotected sex, leading to a high prevalence of unintended pregnancies and unsafe abortions, as well as an increased risk of STIs [14,15]. Studies indicate that contraceptive use among Ghanaian adolescents remains low, despite increasing knowledge about modern contraceptive methods [16–18]. Multiple studies conducted among sexually active adolescents and young women report a high unmet need for modern contraceptive methods [19,20].

One of the necessary approaches to reducing the burden of the consequences of unprotected sex is the use of contraceptives [21]. Contraceptive use is highly effective in preventing unwanted pregnancies, unsafe abortions, and abortion-related complications, which expose adolescents to health-related risks such as infertility and, in some cases death [21]. Globally, contraceptive use in regions such as Europe and North America exceeds 70%, whereas usage in Western and Central Africa remains below 25%, illustrating a wide gap in access and utilization [22].

Many studies on adolescent sexual behaviour in the Ghanaian context have predominantly focused on adolescent girls and young women [17,23–25]. Additionally,

studies that included all adolescents have primarily focused on those in late adolescence [21,26,27], failing to capture the full range of adolescent experiences. Notably, research specifically targeting adolescents in senior high schools is lacking. These students are typically at a developmental stage where sexual experimentation increases [28], yet, their access to credible sexual and reproductive health information is often limited, and in some cases, they are unable to utilize the available information appropriately [28].

Contraceptive use in Ghana remains low, as reported by multiple studies [29–31]. A study by Mavis et al. [29] revealed that 26.7% of late adolescents used contraceptives in a cross-sectional study. Similarly, Begetayinoral et al. [30] reported that 26.3% of women of reproductive age were using modern contraceptives. These findings are below the national acceptor rate, which is 33.8% [32]. In addition, factors influencing contraceptive use among high school adolescents have not been adequately assessed. Addressing this issue is crucial not only for improving adolescent health outcomes but also for advancing economic growth and long-term national development. Understanding the factors that influence contraceptive use among adolescents is therefore essential, as the existing literature provides important insights into the dynamics surrounding adolescent sexual and reproductive health [33].

Some studies have identified a wide range of factors associated with contraceptive use among adolescents [23,33,34]. For instance, Michael et al. [34], in a cross-sectional study among adolescent girls in some sub-Saharan countries, found that girls who had multiple children and prior knowledge of contraceptives were more likely to use them compared to those without such experiences. Similarly, a study by William et al. [33] in Tanzania revealed that factors independently associated with contraceptive use included age, marital status, knowledge of contraceptives, cohabitation, and having multiple sexual partners. However, it is important to note that in the latter study, some participants were interviewed in the presence of their parents, which may have influenced their willingness to respond openly and honestly.

Therefore, given the sparse literature on sexual activity and contraceptive use among adolescents in senior high schools and recognizing that this population represents a critical group with increasing independence, exposure to peer influence, and limited access to reproductive health information within the school setting, this study aims to assess sexual activity and factors influencing contraceptive use within this population. Exploring the sexual behaviour and contraceptive use among this group is vital for informing interventions and policy reforms that will improve sexual and reproductive health.

## 2. Materials and methods

### 2.1. Ethics statement

Prior to conducting the study, an institutional approval was sought from the school's management to conduct the research at the school. Ethical approval, referenced as CHRPE/AP/426/22, was granted by the Committee on Human Research, Publication, and Ethics (CHRPE) at the School of Medicine and Dentistry, Kwame Nkrumah University of Science and Technology in Kumasi, Ghana. Since the majority of participants resided in the school's boarding house and many parents lived far from the school, direct consent from parents was not feasible. Therefore, verbal consent was obtained from the school authorities acting in loco parentis due to the boarding school arrangement for student participation. The participants were briefed on the research's objectives before seeking their consent to participate. They were provided with an opportunity to inquire about the study procedures. Verbal consent was documented by the research team and witnessed by the teachers acting as custodians in the school setting. Participation was entirely voluntary, with no direct benefits to participants, and they were free to withdraw at any point. To protect anonymity, no names or other identifying details were collected.

### 2.2. Study design

The study employed a descriptive cross-sectional study design. This design was deemed appropriate, as the researchers aimed to collect data at a specific point in time and describe the characteristics of adolescents concerning sexual activity and contraceptive use [35].

## 2.3. Study setting

The study was conducted at a senior high school in Ejisu Municipality in the Ashanti region of Ghana, which includes both rural and urban localities [36]. The Ejisu Municipality covers approximately 232.011 km² and is situated in the central part of the Ashanti Region. The municipality is known for its educational institutions, commercial activities, and rich cultural heritage [37].

It is home to over 158 pre-schools, 158 primary schools, 95 junior high schools, 6 senior high schools, 1 private university and 2 vocational institutes [38], reflecting a strong presence of educational infrastructure. This context provided an ideal setting for the study, as it allowed the researchers to access a diverse adolescent population from varying socio-economic and geographical backgrounds.

In the Ghanaian context, senior high school is a second-cycle institution that students attend after graduating from junior high school, typically covering grades equivalent to 10–12 in the international education system. The school is a public mixed-gender institution that admits day students and also has boarding facilities for a portion of the student body. The school offers five (5) programmes; Business, Home Economics, Visual Arts, General Arts and General Science. Facilities within the school include classrooms, administrative offices, a library, laboratories, and outdoor spaces such as playgrounds and sports fields. This school was selected as the study setting due to its access to a student population from diverse socio-economic and cultural backgrounds. The school's status as a large, public, mixed-gender institution offering a wide range of academic programs made it an ideal microcosm for exploring adolescent sexual and reproductive behaviors in the municipality. Additionally, logistical feasibility and the willingness of the school management to approve the study were factors in its selection.

## 2.4. Study population

Between 16th August, 2022 and 15th September, 2022, adolescents of both genders in senior high school were selected for the survey. The actual age range of participants in this study was 11–19 years. However, for the purposes of analysis and discussion, participants were categorized into early adolescence (10–14 years) and late adolescence (15–19 years), based on the WHO classification [39]. SHS 1 students were excluded from the study as they were on vacation during the data collection period. As a result, only SHS 2 and SHS 3 students were included in the study.

## 2.5. Inclusion and exclusion criteria

Adolescents aged 10–19 who were present at school during the study and could speak, understand, and write English were eligible to participate. Students who were on vacation, suspended, critically ill, or had mental impairments were excluded from the study.

## 2.6. Sample size determination and sampling

A sample size of 330 was determined using the Slovin's formula:

$n = \frac{N}{1+N(e)^2}$ [40], and with a confidence interval of 95% (at a Z-score of 1.96), a margin of error of 5%, where **n** signifies the sample size, **N** signifies the population under study and **e** signifies the margin of error (0.05). The total population was 1210 using the total enrollment of students at the time of the study obtained from the school's administrative office. The minimum required sample size obtained was 300. A 10% non-response rate was considered to compensate for a non-response, resulting in a total estimated sample size of 330.

## 2.7. Data collection tool

A structured questionnaire was used in this study, developed based on current literature and tailored to the study context. The questionnaire was pretested to ensure clarity and relevance. It was divided into five sections. The first section focused

on sociodemographic characteristics of adolescents (7 questions). This section included questions on age, gender, class, religion, ethnicity, residential status (i.e., whom the adolescent lives with), and the employment status of the person the adolescent lives with. The second section focused on sexual activity (8 questions). This section covered topics such as whether the respondent had a boyfriend or girlfriend, the age of their sexual partner, whether they had ever had sex, the number of sexual partners, the age at first sexual intercourse, how the first sexual experience occurred (whether they were coaxed, forced, or by their own will), and whether they felt pressured to have sex. The third section focused on awareness of contraceptives (11 questions). Questions in this section included whether the respondent had heard of modern contraceptives, their sources of information, the methods of contraception they were aware of, where one could access contraceptives, and similar inquiries regarding emergency contraceptives. The fourth section examined contraceptive use (11 questions). This section included questions on whether the respondent had ever used contraceptives, which methods they had used, whether they used contraceptives every time they had sex, and the reasons for using them, such as "to avoid teenage pregnancy" or "to prevent STIs." Other questions included whether they had used contraceptives during their first sexual intercourse, where they had accessed contraceptives, and how frequently they had used them, with options like "every time I have sex," "only during my first sexual intercourse," or "once in a while." The fifth and final section focused on attitudes of adolescents toward contraceptives (7 Likert scale items). This section included statements like "I approve of adolescents using contraceptives every time they have sex." Higher scores reflect a positive attitude. Other statements, presented in a negative form and reverse-coded, included: "Using contraceptives before a girl's first birth can lead to infertility," "Sex is not enjoyable when I use a condom," and "People who insist on condom use are promiscuous."

## 2.8. Quality assurance; Validity and reliability

To ensure content validity, we invited five public health experts to assess the relevance of each question to its corresponding factor through a face-to-face approach. This process helped determine whether the questionnaire accurately measured the intended information. Minor adjustments were made to some questions by all five experts to enhance their cultural relevance and acceptability for the study participants.

Following the pretest, the researchers assessed the internal consistency of the scale items measuring adolescents' attitudes toward contraceptive use. The analysis yielded a Cronbach's alpha coefficient of 0.707, indicating acceptable reliability of the scale.

## 2.9. Data collection procedure

A structured questionnaire was administered to 330 adolescents attending a senior high school. A simple random sampling technique was employed to ensure an unbiased selection of participants [41]. The researchers first obtained permission from the school administration, and students were briefed on the purpose of the study before participation. Participants were asked to choose between sealed, opaque, folded papers with an equal number labelled 'yes' and 'no'. Those who selected 'yes' after providing informed consent were included in the study.

To minimize potential bias, teachers were asked to leave the classrooms during the questionnaire administration to ensure participants did not feel intimidated or coerced. The questionnaire, comprising 44 questions, was distributed to participants, and they were given a maximum of one hour to complete it. Data were collected in all classes simultaneously to prevent students from discussing the questionnaire's content beforehand. After collection, the completed questionnaires were reviewed by the research team to ensure completeness and accuracy before analysis.

## 2.10. Measurement of variables

### 2.10.1. Dependent variable. *Contraceptive use.* The use of any method (such as condoms, pills, IUDs, injectables, etc.) to prevent pregnancy and reduce the risk of sexually transmitted infections (STIs) during sexual intercourse. In this study, contraceptive use refers to adolescents who reported using any form of contraception.

 

This was the outcome variable and was measured based on participants' responses to whether they had ever used any form of modern contraceptives (Yes/No). The percentage of participants who responded "Yes" was used to describe prevalence of contraceptive use.

**2.10.2. Independent variables.** Independent variables were the sociodemographic variables (age, sex, class, religion, ethnicity, residential status [the person adolescent lives with], employment status of the person adolescent lives with) and sexual activity characteristics (relationship status, multiple sexual partners), and awareness of contraceptives. These were selected based on multiple existing literatures [17,23,33,34,42].

**2.10.3. Sociodemographic variables.** These included age, sex, class level, religion, ethnicity, residential status, and guardian's employment status. These variables were self-reported by participants and presented as categorical variables

***Sexual Activity.*** refers to the engagement in any form of sexual intercourse (penile-vaginal), by adolescents. In this study, it includes individuals who reported having had sexual intercourse at least once.

Participants self-reported if they had ever engaged in sex. Binary coding (Yes/No) was applied.

***Awareness of Contraceptives.*** This was measured by asking if participants had heard of contraceptives. Participants chose "yes/no" as a response to this question.

***Attitudes toward contraception***: Attitudes toward contraceptive use were assessed using a seven-item tool graded on a five-point Likert-scale (1 = Strongly Disagree, 2 = Disagree, 3 = Neutral, 4 = Agree, and 5 = Strongly Agree). Higher scores indicated more positive attitudes toward contraceptive use. Five of the items were phrased negatively, including "Using contraceptives before a girl's first birth can lead to infertility," "Sex is not enjoyable when I use a condom," and "People who insist on condom use are promiscuous", and were reverse-coded before analysis so that higher scores consistently reflected a positive attitude. The total attitude score was then calculated by summing the responses to all seven items, with a possible range of 7–35. Respondents with total scores above the median were classified as having a positive attitude, while those scoring below the median were considered to have a negative attitude toward contraceptive use.

## 2.11. Data processing and analysis

The researchers confirmed the completeness of all collected questionnaires by reviewing each section, ensuring there were no missing data. Serial numbers were assigned to the questionnaires, which were entered into Microsoft Excel for data cleaning and transferred to Statistical Product and Service Solutions (SPSS) version 22.0 for Windows (IBM SPSS Statistics) for analysis. Descriptive statistics (frequencies and percentages) were used to describe the categorical characteristics of participants.

To identify the factors associated with contraceptive use, a Pearson chi-square test was first conducted to assess the association between the independent variables and the dependent variable. Those found to be significant were further analyzed using binary logistic regression.

In Model 1, univariate binary logistic regression was performed for each significant variable from the Pearson chi-square tests to calculate crude odds ratios (COR).

Model 2 was a multivariable binary logistic regression that included all independent variables (awareness of contraceptives and class level) found to be significant in Model 1, while adjusting for control variables such as age, sex, religion, and residential status, relationship status, and multiple sexual partners. Adjusted odds ratios (AOR) and 95% confidence intervals (CI) were reported. Statistical significance was set at $p \leq 0.05$.

## 3. Results

### 3.1. Sociodemographic characteristics of adolescents

The majority of participants were in their late adolescents stage (n = 290, 87.9%), and were female (n = 196, 59.4%). Most of the participants lived with both parents (n = 194, 58.8%), and were Christian (90.9%, n = 300) (Table 1).

**Table 1. Sociodemographic characteristics of adolescents.**

| Variable | Frequency | Percentage (%) |
|---|---|---|
| **Age** | | |
| 10–14 | 40 | 12.1 |
| 15–19 | 290 | 87.9 |
| **Sex** | | |
| Male | 134 | 40.6 |
| Female | 196 | 59.4 |
| **Class** | | |
| SHS Level 2 | 190 | 57.6 |
| SHS Level 3 | 140 | 42.4 |
| **Religion** | | |
| Christian | 300 | 90.9 |
| Muslim | 30 | 9.1 |
| **Ethnicity** | | |
| Akan | 272 | 82.4 |
| Ga-Adangbe | 12 | 3.6 |
| Ewe | 18 | 5.5 |
| Others (Dagoba, Mamprusi, Fafra) | 28 | 8.5 |
| **Residential status** | | |
| Living with both Parents | 194 | 58.8 |
| Living with one parent | 120 | 36.4 |
| Living alone | 16 | 4.8 |
| **Employment status [of the person adolescent lives with]** | | |
| Unemployed | 129 | 39.1 |
| Employed | 201 | 60.9 |

## 3.2  Sexual activity among adolescents

Half of the participants (n = 166, 50.3%) had a boyfriend or girlfriend, and the majority had multiple sexual partners (n = 64, 50.8%). Additionally, over one-third of participants (n = 126, 38.2%) had ever had sex, and of these, more than half (n = 65, 51.6%) experienced their first sexual intercourse during early adolescence, between the ages of 10 and 14 years. Participants who engaged in sex of their own will were (n = 87, 69%). Furthermore, 119 (36.1%) of the participants reported feeling pressured to have sex, predominantly from friends (n = 70, 58.8%) (Table 2**).**

## 3.3.  Awareness of contraceptives among adolescents

The majority of participants (n = 283, 85.8%) surveyed were aware of contraceptive methods. Study results indicated that 155 (54.8%) were aware of male condoms; the fewest (n = 12, 4.2%) were aware of IUDs. Among those who knew about contraceptive methods, television (n = 144, 50.9%) and peers (n = 113, 39.9%) were the major sources of their information, and the least common were parents (n = 10, 5.4%). Regarding whether modern contraceptives provide 100% protection from pregnancy, a notable number of participants (n = 117, 35.5%) reported that they did not know (Table 3**).**

## 3.4.  Use of contraceptives among adolescents

Among the participants who had ever had sex, 100 (79.4%) had used contraceptives. Condoms (n = 66, 66%) and pills (n = 43, 43%) were the most common methods of contraception. Contraceptive use during the first sexual encounter was

**Table 2.** Sexual activity among adolescents.

| Variable | Frequency | Percentage (%) |
|---|---|---|
| **Having a Boyfriend/Girlfriend** | | |
| Yes | 166 | 50.3 |
| No | 164 | 49.7 |
| **Age of boyfriend/girlfriend (N = 166)** | | |
| 10–19 | 108 | 65.0 |
| More than 19 | 58 | 35.0 |
| **Ever had sex** | | |
| Yes | 126 | 38.2 |
| No | 204 | 61.8 |
| **Do you have multiple sexual partners (N = 126)** | | |
| Yes | 64 | 50.8 |
| No | 62 | 49.2 |
| **Number of sexual partners (N = 126)** | | |
| 1 | 62 | 49.2 |
| 2 | 24 | 19.0 |
| 3 | 12 | 9.6 |
| More than 3 | 28 | 22.2 |
| **Age of first Sexual Intercourse** | | |
| 10–14 | 65 | 51.6 |
| 15–19 | 61 | 48.4 |
| **How first sexual intercourse happened (N = 126)** | | |
| Own will | 84 | 66.7 |
| Coaxed | 17 | 13.5 |
| Forced | 15 | 11.9 |
| **Pressure to have sex** | | |
| Yes | 119 | 36.1 |
| No | 211 | 63.9 |
| **Sources of pressure to have sex (N = 119)** | | |
| Friends | 70 | 58.8 |
| Relatives | 12 | 10.0 |
| Teachers | 10 | 8.4 |
| Boyfriend/Girlfriend | 27 | 22.7 |

reported by 67 (67%). The majority of participants used contraceptives to avoid pregnancy (n = 58, 58.0%). Regarding the frequency of emergency contraceptive use, 20 (42.3%) indicated they use it every time they had sex (Table 4).

### 3.5. Attitude towards contraceptive use

Participants were asked to indicate their level of agreement with seven statements on contraceptive use, using a five-point Likert scale rated as follows; strongly agree = 5, agree = 4, uncertain = 3, disagree = 2, and strongly disagree = 1. All the negatively worded statements were reverse-coded. The minimum, maximum, and median scores were 7, 35 and 18 respectively; participants whose overall scores were above the median score (18) were classified as having a positive overall attitude toward contraceptive use. Overall, 76.7% of the adolescents were found to have a positive attitude towards contraceptive use (Fig 1).

**Table 3. Awareness of contraceptives among adolescents.**

| Variables | Frequency | Percentage (%) |
|---|---|---|
| **Heard of contraceptives (N=330)** | | |
| Yes | 283 | 85.8 |
| No | 47 | 14.2 |
| **Heard of any method of contraceptive(N=330)** | | |
| Yes | 283 | 85.8 |
| No | 47 | 14.2 |
| **Where did you hear it from (N=283)** | | |
| Radio | 105 | 37.1 |
| Television | 144 | 50.9 |
| Peers | 113 | 39.9 |
| Parent | 62 | 21.9 |
| Teacher | 81 | 28.6 |
| Health worker | 79 | 27.9 |
| Internet | 113 | 39.9 |
| **What method of contraception do you know (N=283)** | | |
| Male condom | 155 | 54.8 |
| Abstinence | 35 | 12.4 |
| Implant | 18 | 6.4 |
| Withdrawal method | 44 | 15.5 |
| Female condom | 40 | 14.1 |
| IUD | 12 | 4.2 |
| Urinating after sex | 30 | 10.6 |
| Injectable | 20 | 7.1 |
| Pills | 96 | 33.9 |
| **Heard about emergency contraceptive(N=330)** | | |
| Yes | 185 | 56.1 |
| No | 145 | 43.9 |
| **If yes, where did you hear about this emergency contraceptive from (N=185)** | | |
| Radio | 88 | 47.6 |
| Television | 103 | 55.7 |
| Peers | 74 | 40 |
| Health worker | 73 | 39.5 |
| Teacher | 26 | 14.1 |
| Internet | 73 | 39.5 |
| Parents | 10 | 5.4 |
| **How often do you know emergency contraceptives can be used within a year (N=185)** | | |
| Once a year | 40 | 21.6 |
| Twice a year | 48 | 26.5 |
| Three times a year | 22 | 11.4 |
| More than three times a year | 75 | 40.5 |
| **Do you know where you can access contraceptive (N=330)** | | |
| Yes | 249 | 75.5 |
| No | 40 | 12.1 |
| Unanswered | 41 | 12.42 |

*(Continued)*

**Table 3.** (Continued)

| Variables | Frequency | Percentage (%) |
|---|---|---|
| **If yes, where (N = 249)** | | |
| Pharmacy | 194 | 58.8 |
| Friend | 51 | 15.5 |
| Hospital or clinic | 208 | 63.0 |
| Health personnel | 68 | 20.6 |
| **Can a girl become pregnant from just one sexual intercourse (N = 330)** | | |
| Yes | 240 | 72.7 |
| No | 29 | 8.8 |
| I don't know | 61 | 18.5 |
| **Do contraceptives provide 100% protection from pregnancy (N = 330)** | | |
| Yes | 79 | 23.9 |
| No | 134 | 40.6 |
| I don't know | 117 | 35.5 |

**3.5.1. Responses regarding attitude of adolescents towards contraceptive use.** From the survey, the majority (n = 80, 24.2%) agreed with the use of contraceptives by adolescents. Most participants (n = 129, 39.1%) were uncertain whether the use of contraceptives by a girl before her first birth could lead to infertility. Approximately 117 (35%) of respondents were not certain whether "sex is not enjoyable when I use condom". Participants whose religion was against contraceptive use (n = 141, 42.7%), and 207(62.7%) reported that contraceptive use is not on solely the responsibility of females (Table 5).

### 3.6. Association between socio-demographic characteristics of adolescents and use of contraceptives

The study found that awareness of modern contraceptives ($p = 0.002$) and class level ($p = 0.025$) were statistically significant factors influencing contraceptive use. However, no significant association was found between contraceptive use and certain sociodemographic characteristics, including age, sex, religion, ethnicity, residential status, and the employment status of the person with whom the adolescent lives. Similarly, sexual activity characteristics, such as relationship status and having multiple sexual partners were not significantly associated with contraceptive use (Table 6).

### 3.7. Logistic regression predicting modern contraceptive use among sexually active adolescents

A binary logistic regression analysis was conducted to identify factors associated with modern contraceptive use among sexually active adolescents. Both unadjusted (crude odds ratio, COR) and adjusted odds ratios (AOR) were reported. Age, sex, residential status, religion, relationship status, and number of sexual partners were included as control variables. Results from the univariable analysis revealed that awareness of modern contraceptives was significantly associated with contraceptive use. Adolescents who had heard of modern contraceptives were over seven times more likely to use them than those who had not (COR = 7.360, 95% CI = 1.718–31.524, p = 0.007). Additionally, being in SHS 3 was significantly associated with higher odds of contraceptive use compared to SHS 2 (COR = 2.835, 95% CI = 1.114–7.213, p = 0.029). After adjusting for all covariates in the multivariable model, only awareness of modern contraceptives remained a significant predictor. Sexually active adolescents who had heard of modern contraceptives had approximately seven times higher odds of using them compared to those who had not (AOR = 6.686, 95% CI = 1.515–29.505, $p = 0.012$). Other variables, including age, sex, religion, residential status, relationship status, number of sexual partners, and class level, were not statistically significant predictors in the adjusted model (Table 7).

**Table 4. Use of contraceptive among adolescents who have had sex.**

| Variable | Frequency | Percentage (%) |
|---|---|---|
| **Use of Contraceptives(N=126)** | | |
| Yes | 100 | 79.4 |
| No | 26 | 20.6 |
| **Method of Contraceptive used (N=100)** | | |
| Condom | 66 | 66.0 |
| Pills | 43 | 43.0 |
| IUD | 2 | 2.0 |
| Implants | 3 | 3.0 |
| Injectable | 3 | 3.0 |
| Withdrawal | 14 | 14.0 |
| Safe Period | 23 | 23.0 |
| **Do you use contraceptive anytime you have sex (N=100)** | | |
| Yes | 50 | 50.0 |
| No | 50 | 50.0 |
| **Reasons for using contraceptives (N=100)** | | |
| To avoid teenage pregnancy | 58 | 58.0 |
| To prevent STI | 42 | 42.0 |
| **Discuss Contraceptive use with your partner(N=100)** | | |
| Yes | 59 | 59.0 |
| No | 31 | 31.0 |
| I do not remember | 10 | 10.00 |
| **Use of contraceptive first time you have sex (N=100)** | | |
| Yes | 67 | 67.0 |
| No | 33 | 33.0 |
| **Method of Contraceptive use first time of having sex (N=67)** | | |
| Condom | 46 | 68.7 |
| Pills | 32 | 47.8 |
| Implants | 3 | 4.5 |
| IUD | 1 | 1.5 |
| Withdrawal | 12 | 17.9 |
| Safe Period | 18 | 26.7 |
| Injectable | 1 | 1.5 |
| **Access to Contraceptive(s) (N=67)** | | |
| Hospital | 16 | 23.9 |
| Pharmacy | 49 | 73.1 |
| Friends | 22 | 32.8 |
| Health Personnel | 7 | 10.4 |
| **Frequency of Contraceptive use (N=100)** | | |
| Every time I have sex | 50 | 50.0 |
| Only on my first sexual intercourse | 7 | 7.0 |
| Once a while | 43 | 43.0 |
| **Use of Emergency Contraceptives(N=75)** | | |
| Yes | 47 | 62.7 |
| No | 28 | 37.3 |

*(Continued)*

**Table 4.** (Continued)

| Variable | Frequency | Percentage (%) |
|---|---|---|
| **Frequency of Emergency Contraceptive use (N=47)** | | |
| Every time I have sex | 20 | 42.5 |
| Once a year | 10 | 21.3 |
| Twice a year | 9 | 19.1 |
| Three times a year | 4 | 8.5 |
| More than three times a year | 4 | 8.5 |

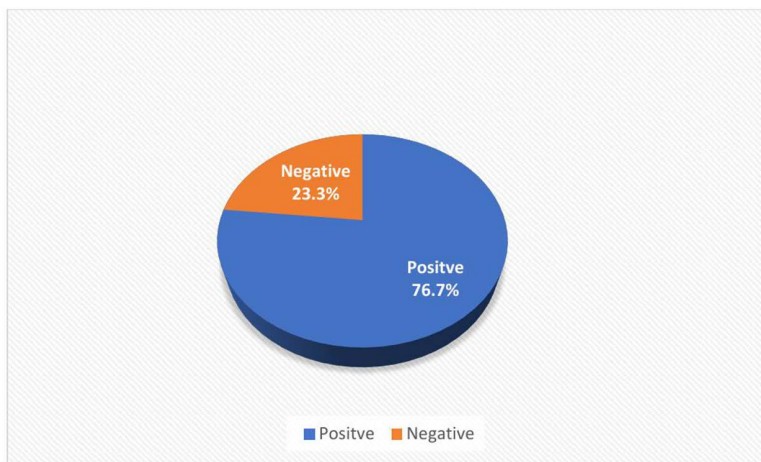

**Fig 1. Attitude of Adolescent towards contraceptive use.**

## 4. Discussion

This study aimed to assess sexual activity and contraceptive use among adolescents. Findings from the study showed that a slightly over half of adolescents were in a relationship with someone of the opposite sex. This corroborates findings from other studies, which have shown that the majority of adolescents have been in a relationship with someone of the opposite sex [39,43].

Furthermore, the study found that among adolescents who had engaged in sexual activity, the majority had their sexual debut during the early adolescence stage (10–14 years). This may be explained by the exploration and heightened sexual curiosity that occur among adolescents during their critical growth phase of identity formation. A study conducted in Kenya reported that adolescents who initiate sex at an early stage are influenced by peer pressure as well as easy access to phones and the internet [44]. This early exposure may be due to increased exposure to erotic materials (nudity, photographs/videos of sexual activity), which are readily available. This increases their curiosity and ultimately leads to engaging in sexual activity. This is further supported by the findings of the current study, which revealed that adolescents mostly feel pressured by their friends and their boyfriend or girlfriend to have sex.

The study also reported a high percentage of adolescents having multiple sexual partners. This is consistent with the findings of Attibu [43] who reported that nearly half of the adolescents in the study had more than one sexual partner. This can be an indicator of increased risk for pregnancy, unsafe abortion, and STIs, suggesting that adolescents may not maintain long-term sexual relationships but instead enter new relationships due to unmet expectations, such as financial rewards or other benefits [45].

**Table 5. Attitude of adolescents towards contraceptive use.**

| Variable | Frequency | | | | |
|---|---|---|---|---|---|
| | Strongly disagree | Disagree | Uncertain | Agree | Strongly agree |
| I approve the use of contraceptive by adolescents anytime they have sex [preferred] | 53(16.1%) | 73(22.1%) | 58(17.6%) | 80(24.2%) | 66(20.0%) |
| Use of contraceptive by a girl before her first birth can lead to infertility | 35(10.6%) | 46(13.9%) | 129(39.1%) | 73(22.1%) | 47(14.2%) |
| Sex is not enjoyable when I use condom | 51(15.5%) | 30(9.1%) | 117(35.3%) | 71(21.5%) | 61(18.5%) |
| People who insist on condom use are promiscuous | 44(13.3%) | 68(20.6%) | 86(26.1%) | 68(20.6%) | 64(19.4%) |
| Contraception usage is against my religion | 59(17.9) | 75(22.7%) | 55(16.7%) | 74(22.4%) | 67(20.3%) |
| I will use contraception in the future | 96(29.1%) | 79(23.9%) | 79(23.9%) | 47(14.2%) | 29(8.8%) |
| Contraception is the responsibility of only females | 125(37.9%) | 82(24.8%) | 51(15.5%) | 42(12.7) | 30(9.1%) |

The findings from this study revealed that awareness of contraceptives was high among adolescents. This is in keeping with a study conducted by Agbanyo [16], which revealed that 88% of adolescents were aware of contraceptives. According to our findings, all the adolescents who had heard about contraceptives were able to mention at least one type of contraceptive and where it could be accessed. This high awareness of contraception may be attributed to recent changes in social media advertising, which includes both video and audio content. Additionally, the growing availability of artificial intelligence platforms may serve as another source of information on contraceptives for adolescents [46].

In this study, the most frequently mentioned sources of information on contraceptives were television, peers, and radio, which is similar to findings from research conducted in Tanzania [47]. Additionally, a study by Hagan and Buxton indicated that the majority (60.0%) of adolescents reported receiving information through radio or television [48]. The least common source of information on contraceptive use was found to be from parents. This is in agreement with a study conducted by Hagan and Buxton, which also revealed that, only 3.3% of adolescents received information on contraceptives from parents [48]. This could be a result of African cultural setting where discussing sexual issues at home is often considered immoral behaviour [49]. As a result, both adolescents and adults may feel uncomfortable discussing topics related to sexuality and contraceptive use. The source of information is a major factor in providing adolescents with correct information, regarding contraceptives and their use. A reliable source of information from health care providers and mass media, provide adolescents with more accurate information than friends [47].

Findings from the current study revealed a high prevalence of contraceptive use among adolescents, which exceeds the Ghana Health Service's national family planning target of 33.8% [32], and is higher than what has been reported in other studies conducted among adolescents [33,34]. This is in agreement with findings from a study by Maness et al., which also showed high usage of contraceptives by adolescents [50]. However, this is inconsistent with other studies that revealed low use of contraceptives among adolescents [33]. For instance, findings from a study by Kinaro et al. [51] revealed that only 8.6% of all adolescents had ever used a contraceptive method. The high rate of contraceptive use in the present study may be attributed to the high level of awareness as observed among participants.

Condoms and Pills were the most commonly used contraceptives by adolescents. Condoms were the most frequently used method of contraception, likely because it is the most popular, easily accessible, affordable, and provide dual protection against teenage pregnancy and sexually transmitted diseases [52]. The majority of adolescents accessed contraceptives from the pharmacies and peers, which is in line with the findings of Kara et al [47]. Adolescents tend to prefer obtaining condoms and pills from pharmacies, likely because pharmacies are more easily accessible in almost every community compared to hospitals. This corroborates a study conducted by Chandra-Mouli and Akwara, which revealed that pharmacies offer adolescents accessible locations, longer operating hours, and most importantly, anonymity in obtaining contraceptives [53]. The study further indicated that the preference for pharmacies was as a result of difficulty in obtaining contraceptives from health facilities due to restrictive laws, health worker bias, or privacy concerns

**Table 6. Association between socio-demographic characteristics of adolescents and use of contraceptives (N = 126).**

| Variables | Usage of contraceptive | | Chi-square | Total |
| --- | --- | --- | --- | --- |
| | | | (p-value) | (N) |
| | Yes, (%) | No, (%) | | |
| *Age* | | | | 126 |
| 14-16 | 63.6 | 36.4 | 0.343 | 11 |
| 17-19 | 76.5 | 23.5 | | 115 |
| *Sex* | | | | 126 |
| Male | 80.9 | 19.1 | 0.273 | 47 |
| Female | 72.2 | 27.8 | | 79 |
| *Class* | | | | 126 |
| SHS 2 | 68.4 | 31.6 | 0.025* | 76 |
| SHS 3 | 86 | 24 | | 50 |
| *Religion* | | | | 126 |
| Christian | 74.8 | 25.2 | 0.659 | 111 |
| Muslim | 80 | 20 | | 15 |
| *Ethnicity* | | | | 126 |
| Akan | 75.2 | 24.8 | 0.565 | 105 |
| Ga-Adangbe | 66.7 | 33.3 | | 6 |
| Ewe | 100 | 0 | | 5 |
| Others (Dagoba, mamprusi, Fafra) | 70 | 30 | | 10 |
| *Residential status* | | | | 126 |
| Both parents | 73.4 | 26.6 | 0.68 | 64 |
| One parent | 75.9 | 24.1 | | 54 |
| Live alone | 87.5 | 12.5 | | 8 |
| *Employment status* | | | | 126 |
| Unemployed | 71.7 | 28.3 | 0.411 | 53 |
| Employed | 78 | 22 | | 73 |
| *Do you have a boyfriend/girlfriend* | | | | 126 |
| Yes | 76.3 | 23.7 | 0.46 | 114 |
| No | 66.7 | 33.3 | | 12 |
| *Multiple sexual partner* | | | | 126 |
| Yes | 79.7 | 20.3 | 0.256 | 64 |
| No | 71 | 29 | | 62 |
| *Heard of modern contraceptive* | | | | 126 |
| Yes | 78.6 | 21.4 | 0.002* | 117 |
| No | 33.3 | 66.7 | | 9 |

Adolescents exhibited positive attitudes towards contraceptive use. The present study reported that the majority of adolescents approved of the statement "use contraceptive anytime they have sex". These findings contrast with those of a previous study by Sam, where the level of awareness and attitude towards contraceptive use among adolescents were found to be very low and only few of them had ever used or were currently using effective contraceptive methods [54].

However, a few participants exhibited negative attitudes towards contraceptive use. For instance, most adolescents in this study supported the idea that contraceptive use by a girl before her first birth can lead to infertility. This is in agreement with a study conducted by Sam, where 67.0% of respondents strongly agreed that a girl using contraceptives before

**Table 7. Logistic regression predicting modern contraceptive use among sexually active adolescents (N = 126).**

| Variable | COR (95%CL) | p-value | AOR (95%CL) | p-value |
|---|---|---|---|---|
| *Age (years)* | | | | |
| 10–14years | 0.537 (0.146-1.974) | 0.349 | 0.839 (0.201-3.494) | 0.809 |
| 15–19years**(ref)** | | – | | – |
| *Sex* | | | | |
| Male | 1.630 (0.678-3.919) | 0.275 | 1.093 (0.367-3.256) | 0.874 |
| Female**(ref)** | | – | | – |
| *Religion* | | | | |
| Christian | 0.741 (0.195-2.818) | 0.660 | 0.737 (0.175-3.099) | 0.677 |
| Muslim**(ref)** | | – | | – |
| *Residential status* | | | | |
| Both parents | 0.395 (0.045-3.450) | 0.401 | 0.713 (0.071-7.117) | 0.773 |
| Single parent | 0.451 (0.051-4.010) | 0.475 | 0.858 (0.084-8.814) | 0.898 |
| Live alone**(ref)** | | – | | – |
| *Heard of modern contraceptive* | | | | |
| Yes | 7.360 (1.718-31.524) | 0.007* | 6.686 (1.515-29.505) | 0.012* |
| No**(ref)** | | – | | – |
| *Do you have multiple partners* | | | | |
| Yes | 1.674 (0.738-3.801) | 0.218 | 1.711 (0.658-4.648) | 0.271 |
| No**(ref)** | | – | | – |
| *Do you have boyfriend/girlfriend* | | | | |
| Yes | 1.611 (0.450-5.769) | 0.464 | 1.144 (0.281-4.648) | 0.851 |
| No**(ref)** | | – | | – |
| *Class* | | | | |
| SHS 3 | 2.835 (1.114-7.213) | 0.029* | 2.531 (0.872-7.352) | 0.088 |
| SHS 2**(ref)** | | – | | – |

Abbreviations and symbols: *, Statistically significant variables (The significance level was set at ≤ 0.05); COR, Crude odds ratio; AOR, Adjusted odds ratio; (ref), reference category; CI, Confidence Interval.

having her first child could cause infertility [54]. This is also consistent with Lauren et al., who found that future fertility was among the main attitudes that tended to be an obstacle to contraceptive use [55]. Adolescents have a misconception about contraceptive use, possibly because much of the information they receive comes from unreliable sources like friends who lack in-depth knowledge. Additionally, concerns may also result from side effects of contraceptives (especially oral pills and emergency contraceptives) on the menstrual cycle of female adolescents.

Findings from several previous studies have indicated that high awareness of contraceptives does not necessarily translate into increased usage, often resulting in low contraceptive use despite widespread knowledge [16,56,57]. However, the present study reveals a contrasting outcome among sexually active adolescents, a high level of awareness was significantly associated with increased contraceptive use. This discrepancy may be explained by the fact that many sexually active adolescents in this study may be more conscious of the risks associated with unprotected sex, including unintended pregnancies, unsafe abortions, school dropout, and sexually transmitted infections. Consequently, adolescents who had heard of modern contraceptives were approximately seven times more likely to use them compared to their counterparts who had not. In addition, all adolescents who had ever used contraceptives indicated that their primary reason was either to prevent unwanted pregnancy or to prevent STIs.

## 5. Conclusion and recommendation

Adolescent sexual activity and contraceptive use are significant issues that require attention and targeted interventions. The high prevalence of multiple sexual partners and early sexual debut among adolescents poses potential risks for unintended pregnancies, unsafe abortions, and sexually transmitted infections (STIs). On a positive note, there is a high level of awareness and positive attitudes toward contraceptive use among adolescents. However, misconceptions persist, potentially due to unreliable sources of information. Given this, there is a need for healthcare providers and other relevant stakeholders to use multidimensional and individualized interventions to improve the reproductive health of adolescents. These interventions must leverage digital platforms, given the current influence of social media and technology, where interactive mobile applications and online counseling can be utilized to provide reliable information and debunk harmful myths about adolescent reproductive health.

Parents and guardians should be encouraged to educate adolescents about sexual activity and contraceptive use. Collaboration between Ministry of Health and the Ministry of Education is essential for organizing regular health seminars to keep adolescents well-informed about issues related to sexuality and contraception. Additionally, there is a need to reconsider incorporating comprehensive sex education starting at the basic school level. Peer educators should be identified and trained to provide sexual education, as adolescents often feel more comfortable seeking information from their peers.

The program of incorporating comprehensive sex education in senior high schools must go beyond biological content to address values, relationships, consent, and communication skills. Emphasis should be placed on peer-led initiatives and youth-friendly approaches that resonate with adolescents' lived experiences while empowering them to make informed choices.

Efforts should be directed towards providing training to health workers and educators on how to effectively communicate accurate, evidence-based information to parents and the general public. This may include workshops on health communication and the use of trusted sources. Finally, future studies should explore the role of parents and guardians in influencing adolescents' sexual activity and contraceptive use, as well as assess the effectiveness of school-based sexual and reproductive health policies.

## 6. Limitations of the study

The use of self-reporting and the school environment may have influenced participants to provide responses that they perceived as socially desirable. In addition, the researchers used a small sample size. However, the findings of the current research have relevant implications for improving adolescent sexual and reproductive health.

## Supporting information

**S1 Text. Data collection tool used to gather participant responses.**
(DOCX)

**S1 Checklist. Completed STROBE checklist ensuring adherence to reporting guidelines.**
(DOCX)

**S1 Data. Raw data in Excel format containing all participant responses.**
(XLSX)

## Acknowledgements

The authors sincerely thank the management of the school for granting us the opportunity to conduct this study in their esteemed institution. We also extend our heartfelt gratitude to the students who participated in this study.

## Author contributions

**Conceptualization:** Peter Boakye, Alberta Yemotsoo Lomotey.

**Formal analysis:** Peter Boakye.

**Investigation:** Peter Boakye, Evans Adaboh, Jacob Tetteh.

**Methodology:** Peter Boakye, Evans Adaboh, Jacob Tetteh.

**Project administration:** Peter Boakye.

**Supervision:** Alberta Yemotsoo Lomotey, Abigail Kusi-Amponsah Diji.

**Validation:** Alberta Yemotsoo Lomotey.

**Visualization:** Peter Boakye.

**Writing – original draft:** Peter Boakye, Evans Adaboh, Jacob Tetteh.

**Writing – review & editing:** Peter Boakye, Evans Adaboh, Alberta Yemotsoo Lomotey, Jacob Tetteh, Abigail Kusi-Amponsah Diji, Victoria Bubunyo Bam.

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
