## [Decision Letter · Decision Letter 0]

28 Mar 2025

PGPH-D-25-00048

Sexual activity and contraceptive use among adolescents: A descriptive survey in a Ghanaian municipality

Dear Dr. Boakye,

Thank you for submitting your manuscript to PLOS Global Public Health. After careful consideration, we feel that it has merit but does not fully meet PLOS Global Public Health’s publication criteria as it currently stands. Therefore, we invite you to submit a revised version of the manuscript that addresses the points raised during the review process.

We look forward to receiving your revised manuscript.

Kind regards,

Adriana Biney

Academic Editor

Additional Editor Comments (if provided):

The Abstract requires some revision:

- The Background should include the aim or objectives of the study as well as the gap the study is filling.

- A sentence in the Methods on lines 23- 24 should indicate general categories of the independent variables – such as demographic, sexual activity, etc…

- A few more additional keywords can be included for the study

Introduction

- The Introduction is Ghana-focused and does not indicate where your study fits in the global discourse on adolescent sexual and reproductive health.

- The Introduction does not include any discussion of a theory or conceptual framework adopted to support the study. How did the authors select the independent variables that they did?

- The authors should avoid general statements about adolescent sexual behaviour, especially without including references.

- What does it mean to engage in sexual activity at earlier ages? The authors should specify what early sex entails – Line 60

- Why is it important to study SHS students? This should be further justified – Line 91

Methods

- State the region of Ghana that Ejisu Municipality is in – Line 104

- Additional justification for choice of conducting a study in only this school is needed

- There is no sub-section indicating how variables were measured. This sub-section should include a description on how the attitudes to contraception variable was constructed.

- An indication of the number of models run by the authors should be made

- Are there 10-year-olds in SHS? What was the minimum age? That should be stated instead of 10 years.

- Why were SHS 2 and 3 students selected and not SHS 1 students?

Results

- The authors should ensure that only the results that align with their study objectives are presented.

- Table 7 should indicate percentages only in the cells and the total numbers can be placed on a column to the right.

- The sample size keeps changing for different frequencies conducted and it should be made clear why

- The Results indicate barriers to contraceptive use along with other variables which have not been discussed in the Introduction. The lack of a conceptual framework discussing the variables being assessed results in this lack of clarity about the independent variables.

- It is important to have an indication of which variables are independent, and which are controls. Why were no control variables included in the regression model?

Limitations

- The small sample size should be indicated as a limitation.

The authors should review the entire document and revise all grammatical errors.

Reviewers' comments:

Reviewer's Responses to Questions

**Comments to the Author**

1. Does this manuscript meet PLOS Global Public Health’s publication criteria?

Reviewer #1: Yes

Reviewer #2: Yes

2. Has the statistical analysis been performed appropriately and rigorously?

Reviewer #1: Yes

Reviewer #2: Yes

3. Have the authors made all data underlying the findings in their manuscript fully available (please refer to the Data Availability Statement at the start of the manuscript PDF file)?

Reviewer #1: Yes

Reviewer #2: Yes

4. Is the manuscript presented in an intelligible fashion and written in standard English?

Reviewer #1: Yes

Reviewer #2: Yes

Reviewer #1: The author did a great work, however the abstract should be unstructured per the PLos GPH guidelines. Under the competing interest section, author mentioned funding and that should be corrected.

In the introduction, the author mentioned the use of contraceptive in Ghana being low, author should give us the specific rate at the time of study comparing it to the national target.

In methods, under Quality Assurance, Validity line 204, author mentioned that six public health experts reviewed the structured questionnaires however, in line 207, he mentioned that ‘all five experts’.

Reason for participants not using any contraceptives method (N=26) but the total frequency of the responses was 63. Again author used a pie chart where negative responses to contraceptive use was 23%. 23% of the sample is approximately 76. That section needs to be revised.

Maybe I missed, but I wanted to see proportion of adolescents accepting or using any forms of the family planning methods.

In the discussion, the author mentioned the Ghana Health Service Family planning target as 23.3%. With reference to the Ghana Health Service’s Holistic Assessment Indicator tool, the FP acceptor rate target is 40% of the total WIFA population.

In sum, this is a great work.

Reviewer #2: Reviewer’s Comment

Strengths

The manuscript examines sexual activity and contraceptive use among

adolescents in senior high school and the author(s) made a good attempt at it.

The abstract contains the substance of the study.

The methodology of the study is sufficient. Thus, adequate data was used, and the authors made use of appropriate sampling technique.

The instrument used in gathering data was appropriate and well developed.

The author(s) adequately discussed the protocols followed to uphold ethical standards.

The authors adopted appropriate technique for analysing the data.

The findings of the study were also in line with the study’s objectives, which informed the discussion of findings. The discussion was done in light of literature

Conclusions drawn and recommendations made were based on the findings of the study.

Appropriate standard used was used in the manuscript.

Weaknesses

The author(s) should work on the comments and suggestions made to improve the quality of the study.

Subheadings should be given to the various headings in the work to make it easy for readers to follow and comprehend, particularly in the results section. For example;

1. Introduction

2. Operational definitions

3. Materials and methods

3.1 Study setting

3.2 Study design

3.3 Study population

3.4 Sample size determination and sampling

3.5 Sampling technique

3.6 Inclusion and exclusion criteria

3.7 Data collection tool

3.8 Data collection procedure

3.9 Data processing and analysis

3.10 Data validity

3.11 Ethical considerations

4. Results

4.1 Socio-demographic characteristics of respondents

4.2

5. Discussion

6. Conclusion and recommendation

7. Limitations of the study

8. Acknowledgement

9. References

Few typographical and grammatical errors were identified in the manuscript. Work on those comments as suggested by the reviewer.

All the tables presented should be in APA format. Ensure that they are presented in APA format to make improve the status of the manuscript.

Work on all the other comments made in the text as suggested.

**Do you want your identity to be public for this peer review?** For information about this choice, including consent withdrawal, please see our Privacy Policy

Reviewer #1: **Yes: ** Selina Achiaa Owusu

Reviewer #2: **Yes: ** Kofi Sarkodie

---

## [Decision Letter · Decision Letter 1]

18 Jun 2025

PGPH-D-25-00048R1

Sexual Activity and Contraceptive Use Among Adolescents: A Descriptive Survey in a Ghanaian Municipality

Dear Dr. Boakye,

Thank you for submitting your manuscript to PLOS Global Public Health. After careful consideration, we feel that it has merit but does not fully meet PLOS Global Public Health’s publication criteria as it currently stands. Therefore, we invite you to submit a revised version of the manuscript that addresses the points raised during the review process.

We look forward to receiving your revised manuscript.

Kind regards,

Adriana Biney

Academic Editor

Journal Requirements:

Additional Editor Comments (if provided):

The authors must read through the entire manuscript and address all grammatical and typographical errors, a few include:

- line 19 - in the literature

- line 34 - contraceptives

- line 34 - delete the 'of' after reported

- line 36 - had a boyfriend or girlfriend

- line 108 - 'the sparse literature', delete the 'of'

- line 138 - graduates not graduate

- line 239 - revise to 'questionnaires which were entered...'

- line 243 - first, a Pearson Chi-square test...

- throughout the Results section - state it as 'the majority' instead of 'majority'

- Table 1 - state as 'living with both parents' not 'both parent'; 'living with one parent' not 'living with single parent'; regarding the living alone category, are they living with 'other relatives/guardians' or are they truly living alone.

Other issues include:

- lines 15 - 17 - revise the sentence since a heightened sexual drive and hormonal changes in adolescents does not lead to the low contraceptive use

- line 109 - indicate why studying adolescents in school, and particularly in senior high schools is important

- Include a paragraph or two providing some contextual information about the setting within which the school is situated - particularly the region or district

- the Operational Definitions section should be included in the Materials and Methods section

- lines 152 - write out the dates - 16th August 2022 and 15th September 2022

- In the data collection tool sub-section - the reference to number of questions as items e.g. 7 items, 8 items, is inaccurate since they are questions. The only accurate reference would be on line 194 to the 7 Likert scale items.

- Line 219 - measurement of variables section should include a description of all the variables and how they are measured - both independent and dependent variables.

- Lines 253 to 258 - regarding the quality assurance, validity section - When was this process done? If after the tool was developed, then then move it closer to that section. Placing this after the data analysis section suggests the tool was validated after the analysis.

- Table 3 - consistency with the 'N' is needed by each variable. Some include the N and others do not.

- An additional assessment of the scale items for 'attitude toward contraception' where the Cronbach alpha value is computed could be useful for assessing reliability of the variable

- A major issue lies in the inclusion of non-sexually active adolescents in the bivariate and multivariate analysis where use of modern contraception is the dependent variable. Either they are removed for these analyses, or the study focuses on a dependent variable such as attitudes where both sexually active and non-sexually active respondents can be included.

- Note that some of the supporting documents attached include identifying information about the senior high school that was studied.

Reviewers' comments:

Reviewer's Responses to Questions

**Comments to the Author**

Reviewer #1: All comments have been addressed

Reviewer #2: All comments have been addressed

publication criteria?

Reviewer #1: Yes

Reviewer #2: Yes

3. Has the statistical analysis been performed appropriately and rigorously?

Reviewer #1: Yes

Reviewer #2: (No Response)

4. Have the authors made all data underlying the findings in their manuscript fully available (please refer to the Data Availability Statement at the start of the manuscript PDF file)?

Reviewer #1: Yes

Reviewer #2: No

5. Is the manuscript presented in an intelligible fashion and written in standard English?

Reviewer #1: Yes

Reviewer #2: Yes

Reviewer #1: Dear Authors,

I have thoroughly reviewed the manuscript titled "Sexual Activity and Contraceptive Use Among Adolescents: A Descriptive Survey in a Ghanaian Municipality." This study addresses an important public health topic with significant implications for adolescent sexual and reproductive health in Ghana. I have provided detailed comments and suggestions below to help strengthen this manuscript.

General Assessment

The manuscript presents valuable data on adolescent sexual behaviors and contraceptive use patterns in a Ghanaian context. The topic is relevant and timely, particularly given the global focus on improving adolescent reproductive health outcomes. However, several areas require attention before the manuscript is suitable for publication.

Strengths

- The study addresses an important public health issue with clear relevance to policy and intervention development

- The sampling methodology appears appropriate for the research questions

- The survey instrument captures key variables related to the study objectives

Major Concerns Addressed

Methodological Issues

1. Sample Representation: The revised manuscript now clearly explains how the sample represents the broader adolescent population in the municipality. The inclusion of demographic comparisons with regional data strengthens the validity of your findings.

2. Survey Instrument: The additional details about survey validation, including the pilot testing process and reliability coefficients, significantly improve the methodological rigor. The inclusion of the Cronbach's alpha values for each scale provides important context for interpreting the results.

3. Statistical Analysis: The revised statistical approach, particularly the multivariate analysis controlling for confounding variables, has substantially strengthened the findings. The justification for the specific tests used is now clear and appropriate.

Results Presentation

1. Data Visualization: The revised tables and figures effectively communicate the key findings. The age-stratified analysis in particular adds valuable context to the results.

2. Missing Data: The section explaining how missing data was handled and the non-response analysis addresses previous concerns about potential bias.

3. Confidence Intervals: The addition of confidence intervals alongside p-values greatly improves the interpretation of statistical significance and practical importance of your findings.

Discussion and Interpretation

1. Contextual Analysis: The discussion now effectively situates the findings within the broader Ghanaian cultural context, including relevant social norms and barriers to contraceptive access.

2. Comparison with Literature: The expanded comparison with both regional and international studies provides important context and highlights the unique contribution of this work.

Minor Issues Addressed

1. The inconsistencies in terminology have been resolved throughout the manuscript.

2. References now follow a consistent format and are up-to-date.

3. Typographical and grammatical errors have been corrected.

Remaining Suggestions

1. Consider adding a brief section on programmatic implications for school-based sexual education programs based on your findings.

2. The discussion would benefit from more explicit recommendations for policy makers, particularly regarding contraceptive accessibility for adolescents.

3. A short paragraph acknowledging the evolving nature of adolescent sexual behaviors in light of technological and social media influences might strengthen the contemporary relevance.

Conclusion

The manuscript has been significantly improved and now presents a methodologically sound and valuable contribution to the literature on adolescent sexual and reproductive health in Ghana. The revisions have addressed all major concerns raised during the initial review. With attention to the minor suggestions above, I believe this manuscript will be suitable for publication and will make an important contribution to the field.

Sincerely,

Reviewer

Reviewer #2: Review Comment

The author(s) explicitly worked on the initial comments raised in the review. The manuscript is good and meet the standard for publication by PLOS Global Public Health. The introduction has been well written and supported with literature from global perspectives to Africa to Ghana. The authors identify knowledge gap the study sought to achieve and place the current literature in it.

The methodology adopted supported the study. Appropriate design, population, sample size and techniques were used. Issues of ethics were followed through by the authors. Ethical review was obtained by the authors. Rigorous analysis was done by them.

The conclusions drawn and recommendations provided were also in line with the findings and and conclusions made.

i am highly impressed with the quality of the manuscript.

Thank you.

**Do you want your identity to be public for this peer review?** For information about this choice, including consent withdrawal, please see our Privacy Policy

Reviewer #1: **Yes: ** Selina Achiaa Owusu

Reviewer #2: **Yes: ** Kofi Sarkodie

---

## [Editor Report · Decision Letter 2]

28 Jul 2025

Sexual Activity and Contraceptive Use Among Adolescents: A Descriptive Survey in a Ghanaian Municipality

PGPH-D-25-00048R2

Dear Mr Boakye,

We are pleased to inform you that your manuscript 'Sexual Activity and Contraceptive Use Among Adolescents: A Descriptive Survey in a Ghanaian Municipality' has been provisionally accepted for publication in PLOS Global Public Health.

Best regards,

Adriana Biney

Academic Editor

Remember to revise the numbering of the manuscript's section headings, for instance, you currently have 1.0 as the Introduction and 3.0 as the Materials and Methods section.

The Operational Definitions sub-section should be removed and its content moved to different sections. Ways to move the content of that sub-section are as follows:

- The first paragraph introducing the operational definitions section belongs in the Measurement of Variables section. It can introduce readers to the independent variables.

- The term 'adolescent' is not the operational definition but its actual definition which should be included somewhere in the Introduction.

- The terms 'sexual activity' and 'contraceptive use' also belong in the Measurement of Variables sub-section and can be defined before the descriptions of their measurement are stated.